# Canine Oral Melanoma: Questioning the Existing Information through a Series of Clinical Cases

**DOI:** 10.3390/vetsci11050226

**Published:** 2024-05-17

**Authors:** Carmen G. Pérez-Santana, Ana A. Jiménez-Alonso, Francisco Rodríguez-Esparragón, Sara Cazorla-Rivero, Enrique Rodríguez Grau-Bassas

**Affiliations:** 1Instituto Universitario de Sanidad Animal y Seguridad Alimentaria (IUSA), Universidad de Las Palmas de Gran Canaria (ULPGC), Arucas, 35400 Las Palmas, Spain; ana.jimenez114@alu.ulpgc.es (A.A.J.-A.); enrique.rodriguez@ulpgc.es (E.R.G.-B.); 2Unidad de Investigación Hospital Universitario de Gran Canaria Dr. Negrín, Las Palmas de Gran Canaria, 35010 Las Palmas, Spain; frodesp@gobiernodecanarias.org (F.R.-E.); scazorla@ull.es (S.C.-R.); 3Fundación Canaria Instituto de Investigación Sanitaria de Canarias (FIISC), Hospital Universitario de Gran Canaria Dr. Negrín, Las Palmas de Gran Canaria, 35010 Las Palmas, Spain; 4Departamento de Medicina Interna, Universidad de La Laguna, Tenerife, 38200 La Laguna, Spain

**Keywords:** cancer, dogs, oral melanoma, oncology, biological behavior, prognosis

## Abstract

**Simple Summary:**

Oral melanomas are the most lethal form of canine melanoma. Although the majority of cases are malignant, a population with well-differentiated and slowly progressive tumors has been identified. Twelve dogs, nine with amelanotic melanomas and three with melanotic melanomas, were evaluated, with demographic details indicating a balanced distribution among various breeds. Lymphadenectomies were conducted, revealing a 16.66% metastatic rate in regional lymph nodes. At the time of surgery, clinical staging identified stages I, II, and III, with most cases having non-infiltrated margins and high mitotic indices. Follow-up revealed local recurrences and metastases, prompting additional surgeries and affecting survival rates. This study reports varying outcomes, with some dogs completing one year without recurrence, while others experienced progressive disease, leading to six oral melanoma-related deaths. The characteristics of melanotic and amelanotic melanoma are observed to study differences between them. Despite evidence of different biological behavior, no aggressiveness differences were found between oral melanotic tumor and oral amelanotic tumor. The absence of evidence that existing treatments lead to improved outcomes for oral melanomas makes it interesting to investigate the biological behavior of melanomas, both melanotic and amelanotic, to better understand their prognosis and discover new therapeutic targets.

**Abstract:**

Twelve dogs with oral malignant melanomas (MM) were evaluated in this study, with demographic details indicating a balanced distribution of gender, age, and weight among various breeds. Tumor locations varied, with diverse surgical procedures being performed, including mandibulectomies and maxillectomies. Lymphadenectomies were conducted, revealing a 16.66% metastatic rate in regional lymph nodes. At the time of surgery, clinical staging identified stages I, II, and III, with most cases having non-infiltrated margins and a high mitotic index. Follow-up revealed local recurrences and metastases, prompting additional surgeries and affecting survival rates. This study reports varying outcomes, with some dogs completing one year without recurrence, while others experienced progressive disease, leading to six oral melanoma-related deaths. The characteristics of melanotic melanoma and amelanotic melanoma are observed in order to study differences between them, the degree of aggressiveness, the mortality rate and the possibility of future therapeutic targets. Although high pigmentation has been correlated with a better outcome, we could not find any significant correlation between survival and achromia. Oral benign melanomas might exist, and this could justify variabilities between stage and survival; however, carefulness is required due to their unpredictable behavior. The findings underscore the complexity of oral melanoma cases and highlight the need for further research on effective management strategies.

## 1. Introduction

Malignant melanoma (MM), a proliferation of atypical melanocytes, is the most frequently documented oral malignant tumor in dogs older than 10 years of age [1]. It represents 30% to 40% of oral tumors, followed by squamous cell carcinoma and fibrosarcoma [1,2,3,4,5,6]. There are many breeds affected by this type of tumor, among which Cocker spaniel, Golden and Labrador retrievers, Scottish terrier, Poodle, Dachshund, Chow-chow, and Boston terrier stand out [1,7] The location of oral melanoma within the cavity can vary, but greater involvement has been observed at the level of the gingiva, lips, and cheeks [5], as well as the tongue and tonsils to a lesser extent [1]. There is variation in the degree of pigmentation, and some tumors are completely unpigmented [6]; dogs bearing amelanotic oral melanoma present a shorter lifespan in comparison to dogs bearing melanotic oral melanoma [8]. While most melanomas are pigmented, amelanotic oral melanomas are noted clinically and have previously been reported [9]. In amelanotic melanoma samples, immunohistochemistry achieves a definitive diagnosis in almost all cases [10,11], and melan-A, melanoma-associated antigen (PNL-2), tyrosine reactive protein (TRP)-1, and TRP-2 are useful markers [12]. For some amelanotic tumors, this immunodiagnostic cocktail may fail to define tumor histogenesis [13]. 

Canine oral melanoma (COM) has aggressive characteristics, being a tumor with high metastatic potential and high local invasiveness [4,5,14]. The most common sites of metastasis include regional lymph nodes and the lungs [1,15]. The percentage of involvement of these locations in metastasis ranges from 30.3% to 74.0% at the level of the regional lymph nodes [14,16], as well as from 14.0% to 92.0% for distant metastatic spread to the lungs and other organs [14]. The cause of death described in the majority of dogs with this tumor is distant metastasis rather than local recurrence [17,18,19,20,21].

The survival times of dogs with malignant melanomas of the lips and oral cavity are reported to be short, ranging from less than 4 months in some studies [22] to 5.7 months [1] and 8 months in other studies [23]. The biological behavior of COM can be predicted based on several features that can be observed in these dogs such as the site of growth, size, and clinical stage [5,14], in addition to histological and immunohistochemical characteristics, including the mitotic index, degree of pigmentation, and nuclear atypia [22,23]. Primary tumor size has been found to be extremely prognostic [24]. The WHO staging scheme for dogs with oral melanoma (Table 1) is based on size and metastasis, including stage I (<2 cm diameter tumor), stage II (2 cm to <4 cm diameter tumor), stage III (4 cm or greater tumor and/or lymph node metastasis), and stage IV (distant metastasis) [24]. 

In general, oral tumors are resected surgically through the complete removal of the tumor with the aim of achieving clean surgical margins, since. depending on the tumor type, it could be an important factor to evaluate [25,26,27]. Wide resection is the most effective modality for the eradication of the primary tumor [23,28]. Local control of COM requires us to perform surgery whenever feasible [4,23]. Treatment with wide margins with curative intent is associated with a long median progression-free interval (PFI) and survival time (ST) [29]. Currently, there is no specific guideline that determines the size of the necessary surgical margins at the macroscopic level, but according to the authors, it is advisable to aim for a minimum of 1.5–2 cm of sound tissue all around the oral melanoma when feasible [30]. Median survival times (MST) for dogs with oral melanoma treated with surgery tend to be approximately 17–18, 5–6 and 3 months with stage I, II and III disease, respectively [24].

Location is the major prognostic factor, and lip, and tongue locations might have a better prognosis compared with other locations in the mouth [31]. Furthermore, it is suggested that soft tissue lesions and rostrally located maxillary or mandibular lesions are associated with increased PFI and ST [29,32,33]. Metastatic disease present at the time of diagnosis carries a poorer prognosis and is negatively associated with ST [29]. 

Although two variables with great potential prognostic value are known, such as clean surgical margins and the location of the tumor within the oral cavity, it has not been possible to determine standards [34].

Genomic instability is a main characteristic of cancer. In addition, there are characteristic transcription profiles and widespread aberrant alternative transcription events. Those processes hallmark cancer types, as well as participate in the tumorigenic process. Next-generation sequencing has helped to evaluate mutations, transcriptome profile aberrant processing, and microRNA profiles. So, concerning those latter issues, it has been widely recognized that the evaluation of microRNAs characterizes cancer predictive, as well as treatment, models [35,36]. However, the global deregulated microRNA expression profile of COM is still understudied. Nevertheless, a recent study has observed that miR-145, miR-365, miR-146a, and miR-425-5p are differentially expressed in COM and healthy samples, suggesting that they may play a role in COM pathogenesis [37]. Also, it has been observed that miR-450b, miR-301a, and miR-223 are downregulated in COM and miR-126, miR-20b, and miR-106a are upregulated [38]. But, future studies are necessary to evaluate others. Transcriptomic aberrations in COM have been studied. The authors found 80 genes that were expressed only in COM compared with healthy tissue [39]. Within this group of genes, three have been identified as the most abundant, which are BGN, CXCL8, and PI3, in canine malignant melanoma. Furthermore, the study reveals the high expression of COL1A1, SPARC, and Vimentin-like differentially expressed genes in COM, while, on the other hand, KRT13, KRT71, and S100A8 were not expressed in the melanoma group. Finally, dog chromosomes 1 and 9 were enriched with downregulated and upregulated genes, respectively [39]. Another study discovered that chromosome 30 in COM was significantly associated with the amelanotic phenotype. This study suggests that there may be different chromosomal aberrations in oral melanotic melanoma and oral amelanotic melanoma. For this reason, emphasizing research using the genetic line to confirm and determine if this chromosomal region indeed contains interesting genes seems to be an encouraging path in the investigation to find the next step to the therapeutic target both in dogs and humans [40].

## 2. Materials and Methods

### 2.1. Canine Samples

Dogs affected by oral melanoma that have been presented at the Veterinary Oncology Service of GICOREC IUSA (Gran Canaria, Spain) of the ULPGC since 2021 with a minimum follow-up of 1 year to 2023 were prospectively considered for this study. The dogs were presented for surgical treatment and treated according to the Good Clinical Practice guidelines for animal clinical studies and approved by the bioethics committee of ULPGC (OEBA-ULPGC 33/2020R1).

### 2.2. Tumor Staging and Treatment

Dog candidates for surgical excision of the primary tumor with or without regional lymphadenectomy and histopathologically confirmed diagnosis of oral melanoma were included. The data collected (Table 2) included breed, sex, age, weight, tumor localization and size, and tumor–node–metastasis (TNM) classification [19].

For both the staging and the evaluation of the general health condition, all dogs underwent a complete clinical examination, a complete blood examination (complete blood count and biochemistry), X-rays of the thorax (three views), and total body computed tomography (CT) when indicated. Dogs were excluded from this study if distant metastasis were detected before surgery. 

The surgical procedure used to resect oral melanoma was considered to have a curative intent, excising a minimum of 1.5 to 2 cm of healthy-looking bone, soft tissues, or both (depending on the tumor location) based on the imaging tests performed (X-rays and/or CT).

Histological data (Table 3) included diagnosis, the evaluation of the excision margins (not infiltrated or infiltrated), the mitotic index (MI) (<4/10 high power fields (HPF) or ≥4/10 HPF), and regional metastasis evaluated at the time of surgery. All regional lymph nodes were aspirated and cytologically evaluated, and the excised regional lymph nodes were sent to the histology laboratory for evaluation.

### 2.3. Patient Monitoring

Tissue samples from dogs with oral melanoma were obtained following surgical resection to study their transcriptome and then compared with the healthy tissue samples from the same dogs. The dogs underwent re-examination every 4 months in a year (Table 4). At each of these re-examinations, clinical examination, blood examination, and chest X-rays were performed in order to detect variations between the samples taken prior to tumor removal and the changes that may occur in these determinations between patients considered cured after at least one year of follow-up and those who develop metastases during the same period. 

These samples will be analyzed in the laboratory via RNA extraction and cytokine expression in another study to be conducted, whose objectives, together with clinical follow-up, will be to investigate prognostic and therapeutic targets. 

### 2.4. Statistical Methods

Categorical variables are presented as medians and percentages where adequate. Fisher’s exact test was employed to compare categorical variables, while Cochran’s Q test was utilized to assess dichotomous variables across multiple time points. Statistical analysis was performed using R Core Team 2022 software, version 4.2 (R Foundation for Statistical Computing, Vienna, Austria).

## 3. Results

### 3.1. Demographics

In this study, twelve dogs were included, consisting of six females (all sprayed) and six males (four castrated and two intact). The dogs had a median age of 13.0 years, with a range from 8 to 17 years. The median weight was 9.9 kg, ranging from 4.8 to 48 kg. The majority of the dogs (11 out of 12, or 91.66%) were of six different pure breeds, while one dog (8.33%) was of mixed breed. The breeds represented included five Yorkshire Terriers, two Beagles, and one each of Presa Canario, Labrador Retriever, American Staffordshire Terrier, and Cocker Spaniel. 

### 3.2. Tumor Location, Clinical Staging and Histological Evaluation

Oral melanoma was localized at the mandible gum in three dogs (21.42%), maxillary gum in one dog (7.14%), mucosa of the cheek in five dogs (35.71%), mucosa of the lip in four dogs (28.57%), and tonsil in one dog (7.14%). The maximum dimension of 13 oral melanomas, measured on the day of surgical excision, was less than 2 cm in 4 cases (30.76%), 2–4 cm in 8 cases (61.53%), and greater than 4 cm in 1 case (7.69%). 

Curative intent surgery consisted of three mandibulectomies, two maxillectomies, and eight cheek/lip en bloc excisions with mucosal reconstruction or skin flap reconstruction or a combination of both.

Mandibular lymphadenectomies were performed in nine dogs: in two cases, it was performed bilaterally, while in seven dogs only ipsilateral mandibular and/or retropharyngeal lymph nodes were removed. In three dogs, the lymph nodes were evaluated using fine needle aspiration and cytological examination only where there was no evidence of regional metastasis. The overall metastatic rate at the level of the regional lymph nodes was 16.66% (2/12), while 58.33% (7/12) presented reactive lymphoid hyperplasia of the removed node. This allowed for establishing the definitive postoperative tumor stage (parameter N of the TNM system). Clinical staging [19] identified three stage I (25%), six stage II (50%), and three stage III (25%) cases. 

Histology of the excision margins identified 11 dogs (91.66%) with non-infiltrated margins and 1 dog (8.33%) with infiltrated margins. The mitotic index was ≥4/10 HPF in 11 dogs (91.66%) and <4/10 HPF in 1 dog (8.33%) (Table 5).

### 3.3. Follow-Up and Statistical Data

Two dogs completed one year of follow-up without recurrence. Progressive disease was reported in nine dogs, of which one had a local recurrence only, five had local recurrence and metastasis to regional lymph nodes, one had both local recurrence and distant metastasis, and two had distant metastases only. At the end of the study, five dogs were still alive, while six of them had died from oral melanoma-related causes. 

Cases that presented local recurrence and/or metastasis to regional lymph nodes underwent a second surgery with a curative intent, prolonging survival to between 2 and 7 more months. 

The survival and disease-free rates at 4, 8, and 12 months are reported in the Table 6. Dogs who were still alive after completing the study and who presented local recurrence and/or metastasis did so regardless of presenting only local recurrence and/or regional and/or distant metastasis. 

All possible comparisons among categorical variables irrespective of temporal consideration were assessed using Fisher’s exact test without finding significant statistical differences. Cochran’s Q test was utilized to compare dichotomous variables across multiple time points, revealing no significant differences for any of the analyzed variables.

## 4. Discussion

Oral melanomas are considered to be the most lethal form of canine melanoma, with a reported median survival time of just 65 days in dogs left untreated [28,41,42,43]. Although the majority of cases are malignant [1,5,15], a population with well-differentiated and slowly progressive tumors arising from the mucous membranes of the lip and oral cavity has been evident [31]. Even though retrospective studies carried out show variability in time and space, which makes the relationship between histology and observed behavior complicated, the question remains regarding the true malignant potential of oral melanomas [41]. 

This prospective study describes 12 dogs with oral melanomas treated with surgery, of which 3 dogs were diagnosed with oral melanoma and 9 were diagnosed with oral amelanotic melanoma, in contrast to the literature, which describes that most melanomas are pigmented, while amelanotic oral melanomas are less noted clinically, as has been previously reported [9]. 

Amelanotic COM produces comparatively less melanin and is considered more aggressive than melanotic COM. Previous studies have revealed differences in cell proliferation, the expression of connexins (gap junction proteins), and outcomes between melanotic COM and amelanotic COM [8,10,12], and it is suggested that amelanotic COM has a higher growth fraction [23] than melanotic COM in dogs. These differentiations could be important for understanding the prognosis between melanotic and amelanotic melanomas. Nevertheless, tumor burden and pigmentation are inconsistent indicators of malignant potential [44]. Despite evidence of different biological behavior, no aggressiveness differences were found between oral melanotic tumor and oral amelanotic tumor. However, most of dogs that died during this study were diagnosed with oral amelanotic melanoma. 

Although high pigmentation is correlated with a better outcome, we could not find any significant correlation between survival and achromia.

There are several studies that show certain breeds with an apparent greater predisposition, such as Cocker spaniels, Poodles, and other dogs with heavily pigmented oral mucosa, which have also been shown to be at an increased risk of developing oral melanoma. However, more recently an over-representation of the Chow-chow, Gloden retriever, Labrador retriever, and Pekingese/Poodle mixed breeds has been reported [1,45,46,47]. This study has an over-representation of the Yorkshire terrier breed (41.66%, 5/12). There are known breed-based predispositions to melanoma [44], but it is not known if prognoses for melanoma differ according to breed. It would be interesting to study if there is a genetic predisposition or if, on the contrary, the higher incidence rate coincides with animals with pigmented mucous membranes. 

Throughout the reported studies [34], no comparable criteria were used for the location of the tumor, making the evaluation of location as a prognostic indicator very difficult. The dogs included in this study underwent surgery under the same conditions and were followed-up with the same guidelines in order to make an objective comparison. Given the limited number of patients in this study and the variability in tumor locations, it would be interesting for future studies to standardize the description of the tumor location in order to make a better comparative diagnosis and evaluate the aggressiveness of the tumor in each location.

There are discrepancies about the correlation between stage and survival [24,48], and in some studies, no data were presented [41]. It has been proposed that the true degree of malignancy of oral melanomas may be less than what their biological behavior suggests, with a study showing only 59% of the 92% of oral melanomas classified as malignant metastasizing or recurring [29,41,48,49,50]. In our study of 12 dogs diagnosed with COM, 66.67% had aggressive biological behavior. It is suggested that oral benign melanomas might exist [36], and this could justify variabilities between stage and survival; however, carefulness is required due to their unpredictable behavior [5]. 

The current WHO classification system is not seen as appropriate to determine prognosis based on the clinical stage of oral melanomas [33]. A modified staging system is proposed that includes tumor volume rather than diameter, tumor location, and the mitotic index, as they were found to be prognostic in several studies, one with 41 dogs [30] and another with 70 dogs [29]. Furthermore, recently, a consensus and guidelines on melanomas in dogs and cats has been published and supports these changes [44]. In the evaluation of the current study, dogs revealed differences to the WHO classification in the prognosis of survival in the different stages and the mitotic index; although primary tumor size has been found to be extremely prognostic [24], in this study, it could not be correlated with the prognosis. Regardless of stage, 11 of 12 dogs had a high mitotic index that was not correlated with ST or tumor size, ranging from 1.3 to 7.7 cm in diameter. 

Dogs who were classified as stage I had an MST of 9.6 months (patient (1) 9 months, patient (4) 12 months, patient (7) 8 months), dogs who were classified as stage II had an MST of 9 months (patient (3) 9 months, patient (5) 12 months, patient (6) 12 months, patient (10) 5 months, patient (11) 8 months, patient (12) 8 months), and dogs who were classified as stage III had an MST of 6.6 months (patient (2) 12 months, patient (8) 2 months, patient (9) 6 months), in contrast to previous studies that predicted higher MSTs in stage I (17–18 months) and lower MSTs in stages II and III (5–6 months and 3 months respectively) [24]. 

Although several studies report that the biological behavior of COM can be predicted [5,14,22,23], even with a comprehensive understanding of all of these factors, there are oral melanomas that have an unpredictable biologic behavior.

Due to the small number of patients, investigations with patient standardization would be necessary and further prospective studies are warranted to confirm these results and determine whether the variables included in the WHO staging system are prognostic. Comparing oral melanotic melanoma and oral amelanotic melanoma is required to confirm these results and understand the aggressiveness that this tumor presents. 

Taking advantage of the clinical case series of oral melanoma obtained for this study, blood and tissue samples were taken for future studies.

Unrelated samples of tumor tissue and healthy skin control tissue, which were subjected to total RNA extraction following standard protocols, were evaluated. High-quality samples have been used for library construction for further analysis. The genetic expression between melanotic tumors and amelanotic tumors of which studies are being carried out is interesting. Additionally, serum samples from COM were collected at baseline, the day of the surgery, and every 4 months until completing a year of follow-up, during which time a clinical examination and chest X-rays were performed. The goal is to correlate serum samples with the stage of the disease, an area insufficiently explored in the current literature. These results will be studied in greater depth in the research to follow.

## 5. Conclusions

The absence of evidence that existing treatments lead to improved outcomes for oral melanomas makes it interesting to investigate the biological behavior of melanomas, both melanotic and amelanotic, to better understand their prognosis and discover new therapeutic targets.

## Figures and Tables

**Table 1 vetsci-11-00226-t001:** The World Health Organization (WHO) TNM-based staging scheme for dogs with oral melanoma [24].

T:	Primary Tumor
	T1 Tumor < 2 cm in diameter
	T2 Tumor 2–4 cm in diameter
	T3 Tumor > 4 cm in diameter
N:	Regional Lymph Nodes
	N0 No evidence of regional node involvement
	N1 Histologic/cytologic evidence of regional node involvement
	N2 Fixed nodes
M:	Distant Metastasis
	M0 No evidence of distant metastasis
	M1 Evidence of distant metastasis

Stage I: T1 N0 M0. Stage II: T2 N0 M0. Stage III: T2 N1 M0 or T3 N0 M0. Stage IV: Any T, Any N, and M1.

**Table 2 vetsci-11-00226-t002:** Data collection.

Pt	Breed	Sex	Age (y)	Weight (kg)	Tumor Location	Size	TNM
1	Beagle	F	17	8.7	Cheek	1.8 cm	Stage I
2	Presa Canario	M	8	48.0	Cheek	7.7 cm	Stage III
3	Labrador Retriever	M	14	26.5	Cheek	2 cm	Stage II
4	Mixed	F	12	11.0	Lip	1.5 cm	Stage I
5	American Staffordshire Terrier	F	10	17.1	Lip	2 cm	Stage II
6	Yorkshire Terrier	M	14	8.8	Mandible	2 cm	Stage II
7	Yorkshire Terrier	F	11	4.8	Maxilla	1.8 cm	Stage I
8	Beagle	F	13	12.0	Cheek	2.4 cm	Stage III
9	Cocker Spaniel	M	13	17.0	Lip, mandible, and tonsil	1.3 and 2.5 cm	Stage III
10	Yorkshire Terrier	M	12	7.8	Between maxilla and cheek	2.8 cm	Stage II
11	Yorkshire Terrier	F	14	5.25	Mandible	2.5 cm	Stage II
12	Yorkshire Terrier	M	14	7.0	Lip	2.1 cm	Stage II

Abbreviations: Pt, patient. F, female. M, male. (Y), years. TNM, tumor–node–metastasis classification.

**Table 3 vetsci-11-00226-t003:** Histological data.

Pt	Diagnosis	Margins	MI	Regional Metastasis
1	Oral amelanotic melanoma	Not infiltrated	≥4/10 HPF	Reactive lymphoid hyperplasia
2	Oral amelanotic melanoma	Not infiltrated	≥4/10 HPF	Reactive lymphoid hyperplasia
3	Oral melanoma	Not infiltrated	≥4/10 HPF	Reactive lymphoid hyperplasia
4	Oral amelanotic melanoma	Not infiltrated	≥4/10 HPF	Reactive lymphoid hyperplasia
5	Oral melanoma	Not infiltrated	≥4/10 HPF	Reactive lymphoid hyperplasia
6	Oral amelanotic melanoma	Not infiltrated	≥4/10 HPF	No nodal excision
7	Oral amelanotic melanoma	Infiltrated	<4/10 HPF	No nodal excision
8	Oral amelanotic melanoma	Not infiltrated	≥4/10 HPF	Left regional lymph node
9	Oral melanoma	Not infiltrated	≥4/10 HPF	Left regional lymph node and tonsil
10	Oral amelanotic melanoma	Not infiltrated	≥4/10 HPF	Reactive lymphoid hyperplasia
11	Oral amelanotic melanoma	Not infiltrated	≥4/10 HPF	Reactive lymphoid hyperplasia
12	Oral amelanotic melanoma	Not infiltrated	≥4/10 HPF	No nodal excision

Abbreviations: Pt, patient. MI, mitotic index. HPF, high-power fields.

**Table 4 vetsci-11-00226-t004:** Follow-up of the dogs enrolled in this study.

Pt	First Examination	Second Examination	Third Examination
1	Local recurrence	Progressive disease	Oral melanoma-related death
2	No recurrence	No recurrence	No recurrence
3	Local recurrence	Progressive disease	Progressive disease
4	No recurrence	Progressive disease	Progressive disease
5	No recurrence	No recurrence	No recurrence
6	No recurrence	No recurrence	Local recurrence
7	No recurrence	Progressive disease	Oral melanoma-related death
8	Oral melanoma-related death	-	-
9	Progressive disease	Oral melanoma-related death	-
10	Local recurrence	Oral melanoma-related death	-
11	No recurrence	Oral melanoma-related death	-
12	No recurrence	No recurrence	Non oral melanoma-related death

Abbreviations: Pt, patient. First examination, follow-up for the first 4 months. Second examination, follow-up up to 8 months. Third examination, follow-up until the end of the year.

**Table 5 vetsci-11-00226-t005:** Clinical and histological characteristics of canine oral malignant melanoma present in this study.

		Overall Population (12) *
Localization (%)	Mandible	3 (21.42%)
	Cranial	1 (7.14%)
	Middle	1 (7.14%)
	Caudal	1 (7.14%)
	Maxilla	1 (7.14%)
	Cranial	0 (0%)
	Middle	0 (0%)
	Caudal	1 (7.14%)
	Cheek	5 (35.71%)
	Upper face	4 (28.57%)
	Lower face	1 (7.14%)
	Lip	4 (28.57%)
	Upper	2 (14.28%)
	Lower	2 (14.28%)
	Tongue	0 (0%)
	Palate	0 (0%)
	Tonsil	1 (7.14%)
Clinical stage (%)	Stage I	3 (25%)
	Stage II	6 (50%)
	Stage III	3 (25%)
	Stage IV	0 (0%)
Margins	Not infiltrated	11 (91.66%)
	Infiltrated	1 (8.33%)
Mitotic index (MI)	≥4/10 HPF	11 (91.66%)
	<4/10 HPF	1 (8.33%)
Regional metastasis	No nodal excision	3 (25%)
	Metastasis	2 (16.66%)
	No metastasis	7 (58.33%)

* There are 12 dogs with a total of 14 oral melanoma tumor sites.

**Table 6 vetsci-11-00226-t006:** DFI rate, disease progression, and survival in months.

	Months	Overall Population (12)
DFI rate		No recurrence
	≤4	7 (58.33%)
	≤8	4 (33.33%)
	≤12	2 (16.66%)
		Local recurrence
	≤4	4 (33.33%)
	≤8	2 (16.66%)
	≤12	1 (8.33%)
		Progressive disease (regional metastasis)
	≤4	2 (16.66%)
	≤8	2 (16.66%)
	≤12	0 (0%)
		Progressive disease (distant metastasis)
	≤4	2 (16.66%)
	≤8	3 (25.00%)
	≤12	0 (0%)
Survival rate		Alive
	≤4	11 (91.66%)
	≤8	8 (66.66%)
	≤12	5 (41.66%)
		Oral melanoma-related death
	≤4	1 (8.33%)
	≤8	3 (25.00%)
	≤12	2 (16.66%)

Abbreviations: DFI, disease-free interval.

## Data Availability

All data contained within this article.

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
