# Peer review of "Canine Oral Melanoma: Questioning the Existing Information through a Series of Clinical Cases"

_vetsci, 2024, doi:10.3390/vetsci11050226_

Round 1

Reviewer 1 Report

Comments and Suggestions for Authors

INTRODUCTION

The introduction provides a well-structured review of canine oral melanoma, highlighting its prevalence, biological characteristics, and impact across different dog breeds. The originality is moderate as the literature on canine melanoma is already well-documented. The paragraph regarding transcription and transcriptome is unnecessary, and it is beyond the results of this paper.

METHODOLOGY

The methodology section details a prospective study design. The inclusion and exclusion criteria, diagnostic approaches and treatments methods are well described. However, the originality in methodology is limited as it follows conventional veterinary clinical study protocols.

The sentence "Dogs that were candidates for surgical excision..." is a bit long and could be made more concise.

The phrase "with that of heal healthy tissue samples" seems to be a typographical error. It should likely be "with that of healthy tissue samples."

The term "Dogs oral melanoma tissue samples" might be better expressed as "Tissue samples from dogs with oral melanoma."

Some sentences could be broken down into smaller ones to enhance readability.

The text outlines the collection of melanoma tissue samples post-surgery for transcriptome analysis and cytokine assay for a future research. This sentence may not contribute tho the reader’s understanding of the current study findings. Therefore, I suggest remove or relegate it to a section dedicated to future research direction.

RESULTS

I recommend some improvements for clarity and formatting: correction of orthographic errors, consistency in describing data, especially numbers and percentages and simplifying text. Follow an example of corrected text (line 122 to line 128) “In this study, twelve dogs were included, consisting of six females (all spayed) and six males (four castrated and two intact). The dogs had a mean age of 12.6 years and a median age of 13.0 years, with a range from 8 to 17 years. The mean weight was 14.49 kg, and the median weight was 9.9 kg, ranging from 4.8 to 48 kg. The majority of the dogs (11 out of 12, or 91.66%) were of six different pure breeds, while one dog (8.33%) was of mixed breed. The breeds represented included five Yorkshire Terriers, two Beagles, and one each of Presa Canario, Labrador Retriever, American Staffordshire Terrier, and Cocker Spaniel

I would like to know the reason for presentation of mean and median? If there is no reason for that, I recommend the median.

The author did not explore the statistical potential of results. There is no proportional statistical test to attest differences among the percentage of tumor location, clinical staging, and histological evaluation.

For example, in figure 4, the author could explore binary logistic regression to identify predictors for binary outcomes, such as the occurrence of progressive disease. A Chi-square Test or Fisher’s Exact Test that could be used for categorical data to investigate if there are nonrandom associations between two variables, for example, recurrence and survival outcome or recurrence and type of tumor. In figure 5, a Fisher’s exact text might be used. This test could be used to assess the association between categorical variables such as localization, clinical stage, Margins, Mitotic index, regional metastasis.

All tables are represented as figures. Is this a recommendation of this journal?

DISCUSSION

The discussion section addresses several important aspects of canine oral melanoma but could be better from further depth and critical analysis.

The discussion about pigmentation as a prognostic factor would be better if the authors discuss possible biological mechanisms behind this observation and its implications for diagnosis and treatment.

The discussion regarding breed predispositions would be better from speculation on whether this change is due to genetic predispositions or biases in the study samples.

The discussion about tumor localization did not contribute for the study. However, this discussion could be improved by suggesting how future studies might standardize tumor location description for better comparative analysis.

The discussion regarding discrepancies in stage and survival correlation would be useful if the author explores more deeply why these discrepancies might exist.

The discussion about WHO classification system should be improved. The authors can include data from other studies that directly support the proposed changes.

Overall, while the discussion provides valuable information about melanoma, it would benefit from a more detailed analysis and a clearer connection between the current findings and the existing body of literature. Integrating more comparative data, suggesting concrete improvements to methodologies, and hypothesizing about underlying biological mechanisms would strengthen the section significantly.

Comments on the Quality of English Language

No comments.

Author Response

Comments 1: The introduction provides a well-structured review of canine oral melanoma, highlighting its prevalence, biological characteristics, and impact across different dog breeds. The originality is moderate as the literature on canine melanoma is already well-documented. The paragraph regarding transcription and transcriptome is unnecessary, and it is beyond the results of this paper.

Response 1: Thank you very much for the reviewing effort and your kind words. We agree with your comment on the paragraph regarding transcription and transcriptomics as it does not fit the presentation of results. However, since we are also suggested to add a paragraph on future research to be carried out and these analyses are part of the future direction of our work, we decided to justify it with this section (line 230-240).

As you comment, the literature on canine melanoma is already well-documented but we found interesting to conduct a prospective study of oral malignant melanoma, standardizing patients and evaluating the prognostic potential of tumor size and location. In addition, we found a high population of oral amelanotic melanomas that allowed us to compare their evolution with respect to oral melanotic melanomas.

Comments 2: The methodology section details a prospective study design. The inclusion and exclusion criteria, diagnostic approaches and treatments methods are well described. However, the originality in methodology is limited as it follows conventional veterinary clinical study protocols.

Response 2: Thank you very much for your comment. We have tried to reproduce and standardize the conditions suggested including variables to analyze the WHO staging system with prospective clinical case studies as recommended previous studies.

Comments 3: The sentence "Dogs that were candidates for surgical excision..." is a bit long and could be made more concise.

Response 3: Accordingly to the reviewer suggestion, we have modified the sentence to “Dogs candidates for surgical excision…” (line 135)

Comments 4: The phrase "with that of heal healthy tissue samples" seems to be a typographical error. It should likely be "with that of healthy tissue samples."

Response 4: We have made the change as suggested by the reviewer. (line 162)

Comments 5: The term "Dogs oral melanoma tissue samples" might be better expressed as "Tissue samples from dogs with oral melanoma."

Response 5: Accordingly to the reviewer suggestion we have made the changes. (line 161)

Comments 6: The text outlines the collection of melanoma tissue samples post-surgery for transcriptome analysis and cytokine assay for future research. This sentence may not contribute to the reader’s understanding of the current study findings. Therefore, I suggest remove or relegate it to a section dedicated to future research direction.

Response 6: In a similar way to what was indicated regarding the transcriptomics studies, it was justified with the future research section. (line 230-240)

Comments 7: I recommend some improvements for clarity and formatting: correction of orthographic errors, consistency in describing data, especially numbers and percentages and simplifying text. Follow an example of corrected text (line 122 to line 128) “In this study, twelve dogs were included, consisting of six females (all spayed) and six males (four castrated and two intact). The dogs had a mean age of 12.6 years and a median age of 13.0 years, with a range from 8 to 17 years. The mean weight was 14.49 kg, and the median weight was 9.9 kg, ranging from 4.8 to 48 kg. The majority of the dogs (11 out of 12, or 91.66%) were of six different pure breeds, while one dog (8.33%) was of mixed breed. The breeds represented included five Yorkshire Terriers, two Beagles, and one each of Presa Canario, Labrador Retriever, American Staffordshire Terrier, and Cocker Spaniel

Response 7: Now the rewritten paragraph (line 183-189) appears as the suggestion proposed.

Comments 8: I would like to know the reason for presentation of mean and median? If there is no reason for that, I recommend the median.

Response 8: We have made the change as suggested by the reviewer.

Comments 9: The author did not explore the statistical potential of results. There is no proportional statistical test to attest differences among the percentage of tumor location, clinical staging, and histological evaluation.

For example, in figure 4, the author could explore binary logistic regression to identify predictors for binary outcomes, such as the occurrence of progressive disease. A Chi-square Test or Fisher’s Exact Test that could be used for categorical data to investigate if there are nonrandom associations between two variables, for example, recurrence and survival outcome or recurrence and type of tumor. In figure 5, a Fisher’s exact text might be used. This test could be used to assess the association between categorical variables such as localization, clinical stage, Margins, Mitotic index, regional metastasis.

Response 9: As the reviewer aptly points out, this issue holds particular significance. To enhance the comprehensiveness of our study, we have incorporated a brief section detailing the statistical methods utilized (Section 2.4 Statistical Methods; lines 175-180)). It is important to acknowledge that the presented tables depict a case series with a limited sample size, predominantly comprising categorical variables pertaining to patients' tumors characterized under similar parameters. Despite these constraints, all conceivable comparisons among categorical variables, irrespective of temporal variations, were subjected to Fisher's exact test, yielding no statistically significant differences. Furthermore, Cochran's Q test was administered to assess dichotomous variables across multiple time points, elucidating the absence of significant differences for any of the variables under scrutiny. These findings have been incorporated into the manuscript, specifically in lines 226-229.

Comments 10: All tables are represented as figures. Is this a recommendation of this journal?

Response 10: According to the reviewer we have replaced figures with tables.

Comments 11: The discussion section addresses several important aspects of canine oral melanoma but could be better from further depth and critical analysis.

Response 11: Thank you very much for the detailed review effort. Accordingly, we made the following changes: A sentence was added in paragraph 3, lines 258 to 266, in paragraph 4, lines 276-280, paragraph 5, lines 284-288, paragraph 6, lines 294-296 and paragraph 9, lines 319-323.

Comments 12: The discussion about pigmentation as a prognostic factor would be better if the authors discuss possible biological mechanisms behind this observation and its implications for diagnosis and treatment.

Response 12: According to the reviewer we have discussed possible biological mechanisms. Sentences added in lines 43-51, and lines 247-250, and lines 258-266, and lines 319-323; as follows:

Lines 43-51: There is variation in the degree of pigmentation and some tumors are completely unpigmented6 and dogs bearing amelanotic oral melanoma present a shorter lifespan in comparison to dogs bearing melanotic oral melanoma8. While most melanomas are pigmented, amelanotic oral melanomas are noted clinically and have been previously reported9. In amelanotic melanoma samples, immunohistochemistry achieves a definitive diagnosis in almost all cases10,11 and melan-A, melanoma-associated antigen (PNL-2), tyrosine reactive protein (TRP)-1 and TRP-2 are useful markers12. For some amelanotic tumors, this immunodiagnostic cocktail may fail to define tumor histogenesis13.

Lines 247-250: Although majority of cases are malignant1,5,15, a population with well-differentiated and slowly progressive tumors arising from the mucous membranes of the lip and oral cavity has been evident31.

Lines 258-266: Amelanotic COM produces comparatively less melanin and is considered more aggressive than melanotic COM. Previous studies have revealed differences in cell proliferation, expression of connexins (gap junction proteins), and outcome between melanotic COM and amelanotic COM8,10,12 and it is suggested that amelanotic COM has a higher growth fraction23 than melanotic COM in dogs. These differentiations could be important to understand the prognosis between melanotic and amelanotic melanomas. Nevertheless, tumor burden and pigmentation are inconsistent indicators of malignant potential44. Despite evidence of different biological behavior, no aggressiveness differences were found between oral melanotic tumor and oral amelanotic tumor.

Lines 319-323: Due to the small number of patients, further prospective studies are warranted to confirm these results and determine whether the included variables in WHO staging system are prognostic. Further prospective studies, by comparing oral melanotic melanoma and oral amelanotic melanoma, are required to confirm these results and understand the aggressiveness that this tumor presents.

Comments 13: The discussion regarding breed predispositions would be better from speculation on whether this change is due to genetic predispositions or biases in the study samples.

Response 13: According to what was suggested by the reviewer, we added a sentence about this possibility that is shown in the lines 276-280 of the revised manuscript as follows:

There are known breed predispositions to melanoma44, but it is not known if prognoses with melanoma differ according to breed. It would be interesting to study if there is a genetic predisposition or if, on the contrary, the higher incidence rate coincides animals with pigmented mucous membranes.

Comments 14: The discussion about tumor localization did not contribute for the study. However, this discussion could be improved by suggesting how future studies might standardize tumor location description for better comparative analysis.

Response 14: We have made the improvements as suggested by the reviewer (line 284-288).

Comments 15: The discussion regarding discrepancies in stage and survival correlation would be useful if the author explores more deeply why these discrepancies might exist.

Response 15: We have made speculation as suggested by the reviewer. (line 294-296)

Comments 16: The discussion about WHO classification system should be improved. The authors can include data from other studies that directly support the proposed changes.

Response 16: We have made the change as suggested by the reviewer as now is shown in lines 300-302.

Comments 17: Overall, while the discussion provides valuable information about melanoma, it would benefit from a more detailed analysis and a clearer connection between the current findings and the existing body of literature. Integrating more comparative data, suggesting concrete improvements to methodologies, and hypothesizing about underlying biological mechanisms would strengthen the section significantly.

Response 17: Thank you very much for your kind comments. 

Reviewer 2 Report

Comments and Suggestions for Authors

The manuscript is an interesting study about the behavior of oral melanoma in dogs. There are some suggestions to improve the manuscript:

-Line 54 to 56: the sentence in this paragraph is a suggestion, please change to the end of the manuscript.

- Line 90 and 106: what was the criteria used to do not the lymphadenectomy in all the patients?

-Please indicate in the methodology, what were the statistical analyses used?

-Line 117: this data is important to refer in this study because the relevance regardless of whether it is being used for another study

- Figure 1: please include the bibliographic reference.  Titles of figure and tables are referred twice, please use the guidelines references for this.

-Line 162: please, delete this title

-Discussion and conclusion:  in the introduction is referred the relevance of the chromosomal aberrations in oral melanotic and amelanotic melanoma and how this study suggests this theory, so where is the data that support this, and what is the conclusion that affirms this hypothesis or not?

Author Response

Comments 1: The manuscript is an interesting study about the behavior of oral melanoma in dogs. There are some suggestions to improve the manuscript:

Response 1: Thank you very much for the reviewing effort and your kind words.

Comments 2: Line 54 to 56: the sentence in this paragraph is a suggestion, please change to the end of the manuscript.

Response 2: Accordingly to the reviewer suggestion we have made moved it to the end of the manuscript (line 319-323).

Comments 2: Line 90 and 106: what was the criteria used to do not the lymphadenectomy in all the patients?

Response 3: Lymph nodes were evaluated using fine needle aspirations. We removed lymph nodes that there were an evidence or suspicion of regional metastasis by this technique. We have done changes explaining better this item (line 155-157).

Comments 3: Please indicate in the methodology, what were the statistical analyses used?

Response 3: To enhance the comprehensiveness of our study, we have incorporated a brief section detailing the statistical methods utilized (Section 2.4 Statistical Methods; lines 175-180)). It is important to acknowledge that the presented tables depict a case series with a limited sample size, predominantly comprising categorical variables pertaining to patients' tumors characterized under similar parameters. Despite these constraints, all conceivable comparisons among categorical variables, irrespective of temporal variations, were subjected to Fisher's exact test, yielding no statistically significant differences. Furthermore, Cochran's Q test was administered to assess dichotomous variables across multiple time points, elucidating the absence of significant differences for any of the variables under scrutiny. These findings have been incorporated into the manuscript, specifically in lines 226-229.

Comments 4: Line 117: this data is important to refer in this study because the relevance regardless of whether it is being used for another study.

Response 4: Accordingly to the reviewer suggestion we have added a paragraph on future research to be carried out and these analyses are part of the future direction of our work (line 230-240).

Comments 5: Figure 1: please include the bibliographic reference.  Titles of figure and tables are referred twice, please use the guidelines references for this.

Response 5: We have made the change as suggested by the reviewer. (line 71)

Comments 6: please, delete this title

Response 6: We have made the change as suggested by the reviewer.

Comments 7: Discussion and conclusion:  in the introduction is referred the relevance of the chromosomal aberrations in oral melanotic and amelanotic melanoma and how this study suggests this theory, so where is the data that support this, and what is the conclusion that affirms this hypothesis or not?

Response 7: We added more information about biological differences between oral amelanotic melanoma and melanotic melanoma (line 258-266), suspicion of the possibility or benign oral melanomas (line 294-296) and the need to study both to better understand their biological behavior and be able to find future effective therapeutic targets (line 319-339)

Round 2

Reviewer 2 Report

Comments and Suggestions for Authors

Thank you for adding the suggestions made.  I only have some additional suggestions for the final manuscript:

Line 93 -96: the sentence in this paragraph is a future direction, please change it to the end or discussion of the manuscript.

Line 231: Please, describe the paragraph in the third person and use it at the end of the manuscript or the discussion how a future direction to better structure the sequence of the manuscript.

Conclusion, Line 325: add the conclusion of this study as it was referred in the abstract.  

Author Response

Thank you for adding the suggestions made.  I only have some additional suggestions for the final manuscript:

Response 1: Thank you very much for the reviewing effort and your kind words. 

Line 93 -96: the sentence in this paragraph is a future direction, please change it to the end or discussion of the manuscript.

Response 2: Accordingly to the reviewer suggestion we have moved it to the end of the manuscript (line 319-323).

Line 231: Please, describe the paragraph in the third person and use it at the end of the manuscript or the discussion how a future direction to better structure the sequence of the manuscript.

Response 3: Accordingly to the reviewer suggestion we have changed the paragraph on future research in the third person (line 326-337).

Conclusion, Line 325: add the conclusion of this study as it was referred in the abstract.  

Response 4: We have made the change as suggested by the reviewer. (lines 340-343)